# Hydra: Sequentially-Dependent Draft Heads for Medusa Decoding

**Zachary Ankner** [1,2,*] **Rishab Parthasarathy**[1,*] **Aniruddha Nrusimha**[1]
**Christopher Rinard**[1] **Jonathan Ragan-Kelley**[1] **William Brandon**[1]
[1]MIT [2]MosaicML

## Abstract

To combat the memory bandwidth-bound nature of autoregressive LLM inference, previous research has proposed the speculative decoding framework. To perform speculative decoding, a small draft model proposes candidate continuations of the input sequence that are then verified in parallel by the base model. One way to specify the draft model, as used in the recent Medusa decoding framework, is as a collection of lightweight heads, called draft heads, that operate on the base model's hidden states. To date, all existing draft heads have been sequentially independent, meaning that they speculate tokens in the candidate continuation independently of any preceding tokens in the candidate continuation. In this work, we propose *Hydra heads*: a sequentially-dependent drop-in replacement for standard draft heads that significantly improves the accuracy of draft head speculation. We further explore the design space of Hydra head training objectives and architectures, and propose a carefully tuned Hydra head recipe, which we call Hydra++, that improves decoding throughput by up to $1.31\times$ and $2.70\times$ compared to Medusa decoding and autoregressive decoding respectively. Overall, Hydra heads are a simple and well-motivated intervention on standard draft heads that significantly improve the end-to-end speed of draft head-based speculative decoding. We make our code publicly available at https://github.com/zankner/Hydra.

## 1 Introduction

As transformer-based large language models (LLMs) have entered widespread deployment, research into improving the inference efficiency of these models has become increasingly important. While LLM pretraining achieves high hardware utilization by operating over the entire input sequence in parallel, the efficiency of LLM inference has traditionally been constrained by the need to generate tokens one by one in sequence. On current GPU hardware, the serial nature of LLM decoding makes it a *memory bandwidth-bound* problem, with throughput limited by the movement of large weight matrices from GPU main memory to local registers. As each generation step requires accessing the entirety of the model's weights, but only involves a comparatively small number of FLOPs (processing just one token for each sequence in the batch), LLM decoding tends to under-utilize the GPU's abundant capacity for floating-point computation.

To mitigate the memory bandwidth bottleneck in sequential LLM decoding, recent research has investigated accelerating LLM inference through *speculative decoding*. Speculative decoding uses a smaller *draft model* to propose a multi-token candidate continuation of the current sequence on each generation step. The original LLM then *verifies* all tokens in the candidate continuation in parallel, appending some subset of them to the sequence and discarding the rest. Because each verification step requires only a single forward pass through the original LLM, but may result in more than one token being appended to the sequence, speculative decoding can accelerate decoding by reducing the amount of weight data movement required per generated token.

---

*Equal contribution. Correspondence to ankner@mit.edu.

A critical component in any application of speculative decoding is the choice of draft model, which must be cheap enough such that the cost of querying it does not negate the efficiency gains from querying the base model in parallel, but accurate enough such that the acceptance rate in the verification step remains high. While the draft models used in speculative decoding have traditionally been stand-alone, independently-trained language models, Stern et al. (2018) and Cai et al. (2024) instead investigate structuring the draft model as a collection of lightweight heads operating on the base model's semantically rich hidden states. We refer to the lightweight heads that operate on the original hidden states of the LLM as *draft heads*. In the draft head paradigm, each draft head is responsible for predicting the identity of a token a fixed number of steps into the future.

All draft heads to date make predictions only as a function of the base model's hidden states from previously verified tokens, making them unaware of earlier tokens in the current candidate continuation. Because of the strong statistical dependencies between neighboring tokens in language, this sequential independence limits the prediction accuracy of existing draft head architectures. In this work, we propose *Hydra heads*: a drop-in *sequentially dependent* alternative to standard draft heads that improves token prediction accuracy and thus end-to-end decoding throughput. To construct sequentially dependent draft heads, we set each head's output to be a function of the candidate continuation so far. This simple design change leads to significantly better speculation quality as compared to standard draft heads, increasing the average candidate continuation acceptance length by up to 0.46 tokens. This improvement in speculation quality corresponds to a significant improvement in decoding speeds, with Hydra head-based decoding achieving up to $1.1\times$ better throughput than Medusa decoding.

In addition to proposing Hydra heads, we further explore the design space of their training objective and architecture. We find that extending the depth of the draft head MLPs, using a teacher distillation objective, and adding an extra transformer decoder layer to better encode the already verified sequence achieves up to $1.31\times$ and $2.70\times$ higher throughput than standard Medusa decoding and regular autoregressive decoding respectively.

Finally, we investigate Hydra and Hydra++ decoding in alternative inference settings. The first setting that we consider is batched inference. The next setting we examine is non-greedy decoding. We show that by using typical acceptance sampling (Cai et al., 2024), a non-distribution-preserving verification criterion, Hydra++ can achieve the same quality generations as non-greedy sampling of the base model while not compromising acceptance length.

**Contributions**   In this work, we present the following contributions:

- We analyze the standard formulation of draft heads and observe that they are sequentially independent during decoding. We propose Hydra heads as a sequentially dependent alternative and show that introducing sequential dependence increases end-to-end decoding throughput by up to $1.10\times$ as compared to Medusa decoding (Section 6.1).

- We explore the design space of Hydra heads to produce a draft head recipe Hydra++ that further increases decoding throughput by up to $1.31\times$ and $2.70\times$ over Medusa decoding and standard autoregressive decoding respectively (Section 3.1, Section 6.1).

- We analyze the performance of Hydra decoding in the batched inference setting, demonstrating that it achieves better throughput than Medusa at all batch sizes evaluated (Section 6.2).

- We demonstrate that sampling using the typical acceptance criterion allows Hydra++ to achieve the same quality as sampling from the base model while preserving the throughput benefits of speculation (Section 6.3).

## 2 Background

**Speculative decoding.** *Speculative decoding* (Stern et al., 2018; Leviathan et al., 2023; Chen et al., 2023) provides a general framework for efficient LLM decoding. Speculative decoding generates text by combining an expensive, high-quality *base model* with a cheaper, lower-quality *draft model*. For each decoding step, the draft model generates one or more *candidate continuations*, each of which extends several tokens into the future. We then use a single forward pass through the base model to *verify* these candidate continuations in parallel based on some verification criterion. The verification process determines which candidate tokens will be appended to the sequence and which will be discarded.

In the simplest form of speculative decoding, the draft model only generates a single candidate continuation on each generation step. Letting $x_{\leq t}$ be the sequence that has been generated so far and fixing some speculation length $K$, we query the joint distribution of the draft model $p_{\text{draft}}(x_{t+1}, \ldots, x_{t+K} \mid x_{\leq t})$ to generate a candidate continuation $\hat{x}_{t+1}, \ldots, \hat{x}_{t+K}$. We then invoke the base model on the candidate continuation to compute the conditional probabilities: $p_{\text{base}}(\hat{x}_{t+1} \mid x_{\leq t}), \ldots, p_{\text{base}}(\hat{x}_{t+K} \mid x_{\leq t}, \hat{x}_{t+1}, \ldots, \hat{x}_{t+K-1})$; querying the base model is done in a single forward pass. These base model probabilities become the input to the verification criterion, which selects some prefix $\hat{x}_{t+1}, \ldots, \hat{x}_{t+n_{\text{accept}}}$ of the candidate continuation to accept, discarding the rest.

Common verification criteria for use with speculative decoding include rejection resampling (Leviathan et al., 2023; Chen et al., 2023), which guarantees that the output distribution matches the base model's distribution, and greedy acceptance (Stern et al., 2018), in which candidate tokens are accepted if they match the base LLM's most likely prediction. For all verification criteria in common use, using a draft model whose predictions more accurately match those of the base model will result in increased average acceptance lengths, and consequently greater decoding throughput.

**Tree decoding.** Speculative decoding can be generalized to settings in which the draft model proposes a *tree* of candidate continuations, rather than a single candidate continuation (Miao et al., 2023; Spector & Re, 2023; Cai et al., 2024). Nodes of this candidate tree correspond to candidate tokens, and the children of a node represent different possible tokens that might follow it in the continuation. Thus, each path along the tree represents a different candidate continuation. To populate a node with $m$ children, we query the draft model for the $m$ most likely tokens that might follow it, conditioned on the sequence generated so far and the candidate continuation defined by the path to the node from the root of the tree. The children at each node are sorted in descending order of conditional probability. Typically, static trees are employed where the structure of the tree is fixed at design time, meaning that the number of children $m$ at each position in the tree does not depend on any runtime data.

After populating the candidate continuation tree using the draft model, we compute the conditional probabilities of all nodes in the tree using a single forward pass through the base model. We query the base model for these conditional probabilities in a single forward pass by packing all of the tree's tokens into a single input sequence, and manipulating the attention mask to ensure that each token can only attend to its parents in the tree. The conditional probabilities obtained from querying the base model can then be used as input to the same verification criteria that are used in the single-candidate setting.

**Lightweight heads as a draft model.** While typically the draft model used in speculative decoding is an independently-trained language model, Stern et al. (2018) define the draft model as a collection of lightweight heads, which we refer to as *draft heads*, that take as input the base model's hidden state. Taking $K$ to be the maximum speculation length, the draft model used by Stern et al. is defined by a collection of small MLPs $f_{\text{draft},1}, \ldots, f_{\text{draft},K}$ responsible for predicting the tokens $1, \ldots, K$ steps into the future. The predictions of these heads are statistically independent of each other; letting $x_{\leq t}$ denote the sequence generated so far, and letting $h_{t-1}$ denote the last-layer hidden state of the token most recently processed

Figure 1: A visualization of generating a candidate continuation using existing draft heads and using our Hydra draft heads. Lines going into a head represent inputs to the draft head. While the only input to existing draft heads is the base model's last-layer hidden state for the most recently processed token, Hydra heads leverage earlier tokens in the candidate continuation as additional inputs.

by the base model, Stern et al. compute their draft predictions on each generation step as

$$p_{\text{draft}}(x_{t+i} \mid x_{\leq t+i-1}) = f_{\text{draft},i}(h_{t-1})$$

Figure 1 provides a visualization of candidate continuation generation using draft heads.

**Medusa decoding.** Medusa decoding (Cai et al., 2024) is a particular configuration of the techniques listed above. Specifically, it is speculative decoding with a tree of candidates, where the draft model is a collection of draft heads.

While Medusa decoding is agnostic to the architecture used for each draft head $f_{\text{draft},i}$, Cai et al. (2024) choose to use a single-layer MLP with a residual connection.

## 3 Hydra Heads

The key observation behind Hydra heads is that there is no sequential dependence in standard draft heads, i.e., each draft head makes predictions independently. A draft model defined by a collection of draft heads predicts the identity of the $i^{th}$ future token as $f_{\text{draft},i}(h_{t-1})$. However, $h_{t-1}$ is only a function of the already generated sequence $x_{\leq t-1}$. Thus, when using draft heads:

$$p_{\text{draft}}(\hat{x}_{t+i}|x_{\leq t}, \hat{x}_{t+1}, \ldots, \hat{x}_{t+i-1}) = p_{\text{draft}}(\hat{x}_{t+i}|x_{\leq t-1})$$

Intuitively, this means that there is no sequential dependence between draft heads: when we use a draft head to speculate the $i^{th}$ token in a candidate continuation, it is unaware of the $1^{st}, 2^{nd}, ..., (i-1)^{th}$ tokens in the candidate continuation.

We propose Hydra heads, which are sequentially-dependent draft heads. Hydra heads are sequentially dependent as they are a function of both the base model's hidden state up to time $t$ as well as the input embeddings of the tokens sampled by previous Hydra heads. Namely, the draft model is now a collection of Hydra heads $\{f_{\text{Hydra},1}, ..., f_{\text{Hydra},K}\}$ and the $i^{th}$ future token's distribution is parameterized by this collection of Hydra heads as:

$$p_{\text{draft}}(\hat{x}_{t+i}|x_{\leq t}, \hat{x}_{t+1}, \ldots, \hat{x}_{t+i-1}) = f_{\text{Hydra},i}(h_{t-1}, x_t, \hat{x}_{t+1}, ..., \hat{x}_{t+i-1})$$

where $h_{t-1}$ is again the base model's hidden state of the final token in the already decoded sequence. The sequential dependence of Hydra heads v.s. standard draft heads is visualized in Figure 1. We use the term *Hydra Decoding* to refer to speculative decoding with tree candidates and Hydra heads.

As with Medusa, the framework of Hydra decoding is compatible with any choice of model architecture used to implement $f_{\text{Hydra},i}$. The most basic Hydra head architecture we examine is simply a single hidden layer MLP whose input is the hidden state $h_{t-1}$ concatenated with the input embeddings of the preceding tokens in the candidate continuation $E_{x_t}, E_{\hat{x}_{t+1}}, ..., E_{\hat{x}_{t+i-1}}$, where the concatenation is performed along the feature dimension.

### 3.1 Hydra++

With the goal of further improving draft heads, we investigate draft head training objective and architecture improvements orthogonal to the introduction of sequential dependence. We detail this search for the optimal draft head recipe in Appendix A. Ultimately, we find that three changes are beneficial:

1. **Scaling:** We extend the MLP of each head to 4 layers. In our experiments we found that scaling to 5 layers and beyond provides no additional benefit.

2. **Distillation:** Following (Zhou et al., 2024), we train on a self-distillation objective where the draft heads are trained to predict the base model's distribution for a given token instead of the true token.

3. **Prefix Attention:** To improve our draft model's ability to condition on information from across the entire context, as opposed to just the most recently-verified token, we extend the base model with an additional self-attention decoder layer whose only role is to produce more informative hidden states for use as input to the draft model. This added layer is only queried once per decoding step.

We refer to the version of Hydra that leverages all of the above changes as Hydra++.

## 4 Discovering performant decoding trees

Similarly to Medusa, we always perform tree-based speculative decoding with a static tree topology computed offline. Computing a performant tree topology for a given inference setting is nontrivial, because different settings call for different tree topologies; the choice of base model, draft model, batch size (discussed more in Section 6.2), and hardware can all affect the relative performance of different trees.

We derive our decoding trees in a data-driven manner using a two-stage algorithm: first, we find a sequence of "proposal" trees $T_1, \ldots, T_N$ with sizes $1, \ldots, N$ such that each proposal tree approximately maximizes expected acceptance length given its size; then, we choose the optimal tree size for a given setup by empirically measuring the end-to-end throughput achieved using each $T_i$, and selecting the tree which maximizes throughput.

To determine the sequence of proposal trees $T_1, \ldots, T_N$, we follow a simple iterative greedy procedure. We first initialize $T_1$ to the trivial one-node tree. Then, on each step $i$, we simulate speculative decoding using $T_{i-1}$ on a corpus of sample text, and identify the child of an existing node that would yield the greatest improvement in expected acceptance length if added to the tree. We then add this node to $T_{i-1}$ to form the tree $T_i$, and repeat.

After computing $T_1, \ldots, T_N$, we measure the throughput of speculative decoding using each $T_i$ in our desired inference configuration (i.e., batch size, hardware, etc.), and select the tree which empirically maximizes decoding throughput.

In practice, we set the maximum tree size as $N = 100$, and use a 100-question subset of the Alpaca dataset (Taori et al., 2023) to gather our simulated acceptance length and throughput statistics. We provide more details on the trees discovered for each decoding strategy and batch size in Appendix B.

## 5 Shared training and evaluation details

In this section, we detail the elements of our training and evaluation procedure that are common across all our experiments.

Figure 2: Performance comparison on MT-Bench of Hydra++, Hydra, Medusa, and the baseline of autogressive decoding. Hydra heads increase decoding throughput and average acceptance length compared to all other methods.

**Models.** As our base model we build on the Vicuna family of models (Chiang et al., 2023), which are conversation-finetuned LLaMa models (Touvron et al., 2023). We consider 7B, 13B, and 33B parameter Vicuna models.

**Training.** While draft heads can be trained in conjunction with the base model, in this work we only study base models with frozen weights. All models are trained on the ShareGPT dataset (ShareGPT, 2023), a collection of multi-turn conversations. Training is performed on 8× NVIDIA A100-80GB GPUs and conducted using the HuggingFace Trainer (HuggingFace). We use a cosine learning rate schedule with warmup (Loshchilov & Hutter, 2017) and a peak learning rate of 1e-3, and we use the AdamW optimizer (Loshchilov & Hutter, 2019) with parameters $\beta_1 = 0.9, \beta_2 = 0.999$. All Hydra and Medusa heads are trained for one epoch, as we observed that performance for those models saturates at one epoch and fails to improve with further training. All Hydra++ heads are trained for ten epochs.

**Evaluation.** All evaluations are performed on MT-Bench (Zheng et al., 2023), a multi-turn conversation benchmark. Unless otherwise specified, experiments are conducted using speculative decoding with the greedy verification criterion; since there is no stochasticity in the greedy sampling procedure, we do not report the quality of generations as they are identical to the base model. Instead, we report the average *throughput*, which is the number of tokens generated per second, and the average *acceptance length*, which is the number of tokens generated per decoding step, to evaluate the speed and quality of Hydra decoding. We benchmark all 7B and 13B parameter experiments on a single A100-40GB GPU and all 33B parameter experiments on a single A100-80GB GPU.

# 6 Results

In this section we investigate the performance characteristics of Hydra decoding. We examine the effect of draft model architecture on throughput and latency across a range of batch sizes, and also investigate the effect of draft model architecture on generation quality when decoding with non-distribution-preserving verification criteria.

## 6.1 Batch-size-1 decoding throughput experiments

To assess the effect of our interventions on draft model prediction accuracy (and thus decoding throughput), we compare the batch-size-1 decoding throughput achievable using Medusa draft heads, our basic Hydra draft heads, and our enhanced Hydra++ draft heads.

**Effect of Batch Size on Hydra and Medusa**

Figure 3: Performance comparison on MT-Bench of Hydra++, Hydra, Medusa, and the baseline of autogressive decoding for batched inference. Hydra heads increase decoding throughput for all batch sizes examined.

We also include the throughput of non-speculative autoregressive decoding as a baseline. We summarize the results of these decoding throughput experiments in Figure 2.

We find that across all base model sizes evaluated, Hydra achieves higher average acceptance lengths than Medusa, and Hydra++ achieves higher acceptance lengths than Hydra, leading to significant improvements in decoding throughput in both cases. Specifically, Hydra heads achieve a $2.36\times, 2.17\times$, and $2.15\times$ improvement in throughput as compared to autoregressive decoding for the 7B, 13B, and 33B parameter base models respectively. This translates to a throughput improvement over Medusa decoding of $1.11\times, 1.10\times$, and $1.11\times$ for the 7B, 13B, and 33B parameter base models respectively. Furthermore, Hydra++ is even more performant and achieves throughput improvements compared to autoregressive decoding of $2.70\times, 2.50\times$, and $2.53\times$ which translates to throughput improvements over Medusa of $1.27\times, 1.27\times$, and $1.31\times$ for the 7B, 13B, and 33B parameter base models respectively. These results demonstrate that making draft heads sequentially dependent significantly improves their prediction accuracy, and thus their decoding speed. Moreover, these results show that any overheads introduced by passing from Hydra to the more expressive Hydra++ architecture are more than compensated for by the increase in accuracy that those changes enable. We further investigate draft head overheads in Appendix D.

## 6.2 Speculative decoding for batched inference

Speculative decoding techniques are typically evaluated in the batch-size-1 setting. When there is only a single sequence in the batch, decoding is extremely memory bandwidth-bound, and large numbers of FLOPs can be consumed in the verification step of speculative decoding "for free" without significantly increasing the latency per decoding step. However, at larger batch sizes it is easier for verification to become compute-bound, and the number of tokens verified per sequence per step must be more tightly controlled to avoid saturating the GPU's compute capacity and entering the regime where speculative decoding becomes unprofitable.

To assess whether or not the performance gains from Hydra and Hydra++ observed at batch size 1 continue to hold in the batched inference regime, we evaluated the throughput and latency of Medusa, Hydra, and Hydra ++ at batch sizes $\{1,2,4,8\}$ using greedy verification and a 7B base model. We derived the decoding tree used for each draft model and batch size configuration using the algorithm described in Section 4.

**Results** We plot the relationship between batch size, throughput, and latency using the 7B base model in Figure 3. While we find that all speculative decoding techniques outperform standard autoregressive decoding for all batch sizes examined, the relative improvement of speculative decoding decreases as the batch size increases. Specifically, for batch size 1

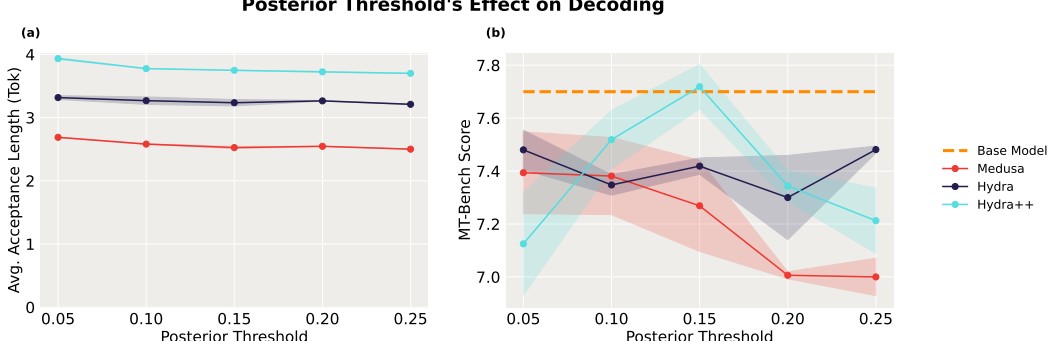

Figure 4: Average speculation length and MT-bench score as a function of the posterior threshold for typical acceptance-based decoding. Hydra++ achieves comparable generation quality to the base model while preserving acceptance length.

Hydra++ has a $2.70\times$ improvement in throughput compared to standard decoding, but this gain decreases to $1.63\times$ at batch size 8. These results suggest that while the gain from Hydra decoding is less significant at larger batch sizes, Hydra decoding is still an improvement over both Medusa and standard decoding at larger batch sizes.

### 6.3 Typical acceptance sampling

So far we have used the greedy acceptance criterion, where candidate tokens are only accepted if they match the greedy next-token prediction of the base LLM. We now evaluate the impact of Hydra decoding on throughput and quality using the non-greedy, non-distribution-preserving *typical acceptance* verification criterion introduced by Cai et al. (2024).

**Typical acceptance criterion.** The purpose of the typical acceptance verification criterion is to sample more diverse and creative sequences than greedy acceptance, while preserving the efficiency benefits of speculative decoding by avoiding the degradation in acceptance rate observed when employing rejection resampling (Gante, 2023; Spector & Re, 2023).

The typical acceptance criterion specifies that a speculated token $\hat{x}_{t+i}$ is accepted if:

$$p_{\text{base}}(\hat{x}_{t+i}|x_{\leq t}, \hat{x}_{t+1}, \ldots, \hat{x}_{t+i-1}; \tau) > \min(\epsilon, \alpha \exp(-H(p_{\text{base}}(\cdot|x_{\leq t}, \hat{x}_{t+1}, \ldots, \hat{x}_{t+i-1}; \tau))))$$

where $\epsilon$ is known as the *posterior threshold*, $\alpha$ is known as the *posterior alpha*, $\tau$ is the sampling temperature, and $H(\cdot)$ is the entropy. Both $\epsilon$ and $\alpha$ are hyperparameters to be tuned. For further analysis of typical acceptance, we refer the reader to Cai et al. (2024).

**Setup.** We evaluate how different settings of $\epsilon$ and $\alpha$ affect both acceptance length and generation quality. Following Cai et al. (2024), we evaluate typical acceptance on the "Writing" and "Roleplay" categories of MT-Bench, and report the average LLM-as-a-judge score to quantify generation quality (Zheng et al., 2023). We fix the sampling temperature $\tau = 0.7$, vary the posterior threshold $\epsilon \in \{0.05, 0.1, 0.15, 0.2, 0.25\}$, and set the posterior alpha as $\alpha = \sqrt{\epsilon}$.

**Results.** We plot how varying the posterior threshold affects both the average speculation length and the quality of the resulting generations in Figure 4. For Medusa, Hydra, and Hydra++, increasing the posterior threshold slightly decreases the average speculation length, but for all posterior thresholds examined, Hydra and Hydra++ have a significantly higher average acceptance length than Medusa. While neither Medusa nor Hydra is able to achieve the same quality as random sampling from the base model for any of the posterior thresholds considered, for $\epsilon = 0.15$ Hydra++ achieves the same generation quality as sampling directly from the base model. These results demonstrate that the improved head quality achieved from Hydra++ is necessary to match the generation quality of the baseline for non-greedy inference while still maintaining a high average acceptance length.

# 7  Related Work

Accelerating LLM inference is an area of active research. The technique our work is based on, speculative decoding, was first proposed by Leviathan et al. (2023) and Chen et al. (2023), and anticipated in a restricted form by Stern et al. (2018). Recent work has explored alternatives to standard draft models such as using retrieval mechanisms to propose continuations (He et al., 2023), and reformulating language model sampling in terms of Jacobi iteration (Fu et al., 2023). Another direction of speculative decoding research has investigated verifying a tree of candidate continuations rather than a single continuation (Miao et al., 2023; Spector & Re, 2023; Cai et al., 2024). In addition to tree decoding, Spector & Re (2023) also propose extending the basic speculative decoding framework by constructing a hierarchy of draft models, with each aiding speculative decoding for the next. Other contemporary directions of research on speculative decoding include online training of the draft model based on user queries (Liu et al., 2023a) and knowledge distilattion based alignment of the draft model (Zhou et al., 2024). We would also like to acknowledge the concurrent work EAGLE (Li et al., 2024) which is the work most similar to ours. We discuss EAGLE in Appendix C. We would also like to acknowledge Zhang et al. (2024); Wertheimer et al. (2024) who concurrently investigated sequentially dependent draft heads.

Another direction for accelerating LLM inference is minimizing the memory impact of LLMs. A common technique is to compress the LLM either by quantizing its weights or pruning the features of the model (Dettmers et al., 2022; Xiao et al., 2023; Frantar et al., 2023; Frantar & Alistarh, 2023; Liu et al., 2023b; Alizadeh et al., 2024; Sheng et al., 2023). To decrease the memory footprint of the KV-cache, Shazeer (2019) and Ainslie et al. (2023) introduce multi-query and grouped-query attention respectively. These works reduce the size of the KV-cache by using fewer key and value heads as compared to the number of query heads in attention. Another method for decreasing the memory footprint of LLMs is knowledge distillation, where a smaller student network is trained to be as accurate as the original larger model (Sanh et al., 2020). These memory-reduction and inference acceleration techniques are orthogonal to, and potentially complementary with, speculative decoding.

Increasing the batch size at which inference is performed is another technique for improving LLM inference throughput. Multiple works investigate better scheduling for batched inference and improved management of shared resources during batched inference (Yu et al., 2022; Kwon et al., 2023).

# 8  Conclusion

In this work, we systematically examine draft head-based speculative decoding and propose methods for improving the speculation quality of draft heads. We make the observation that previously-proposed draft heads are sequentially independent, leading to poor prediction quality. To fix this problem, we propose Hydra heads: a drop-in, sequentially-dependent replacement for standard draft heads. Hydra heads are made sequentially dependent by taking as input the base model's input embeddings of tokens in the candidate continuation. This simple change leads to significant improvements in decoding speed: Hydra decoding achieves up to a $1.11\times$ improvement in end-to-end throughput as compared to Medusa decoding. We also investigate different training objectives and architectures for Hydra heads, ultimately proposing a Hydra head recipe we call *Hydra++* that increases decoding throughput by up to $1.31\times$ and $2.70\times$ as compared to Medusa and autoregressive decoding respectively. Finally, we demonstrate that Hydra++ continues to confer benefits when performing batched inference, and that by using typical acceptance sampling Hydra++ can achieve the same quality as non-greedy sampling of the base model without compromising accepted continuation length. Draft head based speculative decoding is an efficient and simple alternative to the standard speculative decoding paradigm, and our work takes an important step towards maximizing the performance of decoding with draft heads through the construction of sequentially dependent draft heads.

## Acknowledgments

This work was initially performed as a class project for MIT's NLP class, 6.8611, and as such we would like to thank the 6.8611 teaching staff for their collective help. In particular, we would like to thank Jacob Andreas, Yoon Kim, and Chris Tanner for teaching the course, and Marco Nocito and Michael Maune for their feedback on the paper throughout the course.

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

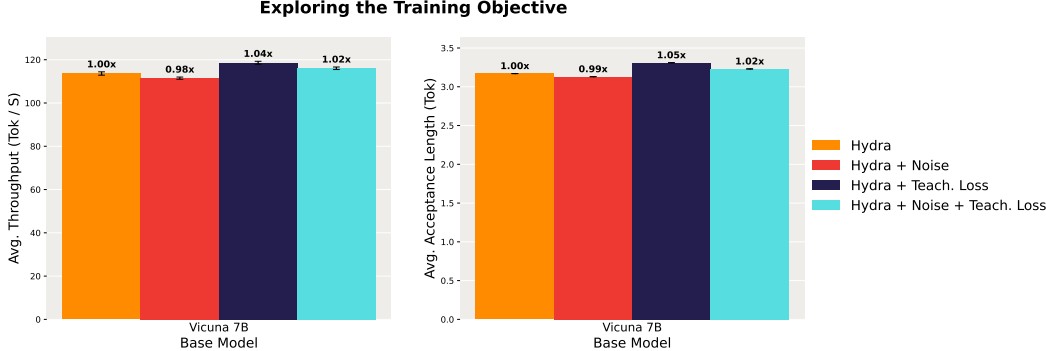

Figure 5: Performance comparison on MT-Bench of different Hydra head training objectives. Training based on a teacher loss leads to the most performant Hydra heads.

## A  Exploring the Design Space of Hydra Heads

In this section we explore modifications to the training procedure and architecture of Hydra heads.

### A.1  Exploring the Training Procedure of Hydra Heads

**Adding noise to the input sequence.**   Jain et al. (2024) demonstrate that adding noise to the input embeddings of an LLM during finetuning can improve the resulting model's performance. Specifically, they sample noise $\epsilon \in \mathbb{R}^{B \times L \times d} \sim \text{Uniform}(-1, 1)$, scale it by a factor $\frac{\alpha_{\text{noise}}}{\sqrt{Ld}}$, and then add the scaled noise to the input embeddings, where $B$ is the batch size, $L$ is the sequence length, $d$ is the model dimension, and $\alpha_{\text{noise}}$ is a hyperparameter that controls the strength of the noise. We consider whether applying such noise to the hidden states of the base LLM can also improve the performance of the Hydra heads. For our experiments, we set $\alpha_{\text{noise}} = 75$.

**Distilling the base LLM.**   In the standard Medusa decoding framework, the draft heads are trained to predict the text of the underlying fine-tuning dataset. We question whether this is the optimal training objective as, during inference, the goal of the draft heads is only to predict the token which the base LLM would have autoregressively predicted. Following Zhou et al. (2024), we investigate using a *teacher loss* where each Hydra head's training loss is the cross entropy between its predicted distribution and the base model's next token distribution.

### A.1.1  Results

To test how each of our training interventions affects Hydra head quality, we evaluate both interventions separately as well as jointly. We compare decoding using Hydra heads trained with the proposed interventions and decoding using Hydra heads trained in the standard manner. We report the decoding throughput as well as the average acceptance length for each intervention in Figure 5. We find that the most performant intervention is to just train on the teacher loss without any additional embedding noise. Specifically, decoding with heads trained with just the teacher loss achieves a $1.04\times$ improvement in throughput for Vicuna 7B compared to decoding with vanilla Hydra heads. Interestingly, we find that any addition of noise to the input sequence degrades the acceptance length and thus the decoding speed. Based on these result, we conduct all future experiments using the teacher loss instead of the next token prediction loss.

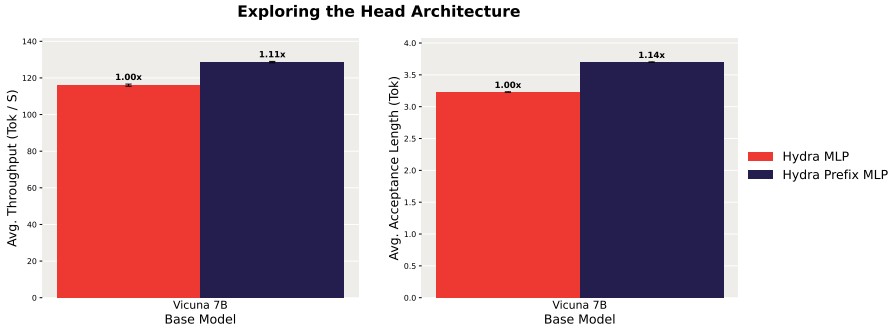

Figure 6: Performance comparison on MT-Bench of the standard MLP only Hydra head architecture and the PrefixMLP head architecture, which introduces an additional decoder layer. The PrefixMLP Hydra head architecture outperforms a standalone MLP Hydra head.

## A.2 Exploring Alternative Hydra Head Architectures

**Hydra-specific prefix attention.** For MLP-based Hydra heads, the only representation of the already-generated sequence passed to the heads as input is the base model's hidden state corresponding to the most recently processed token. However, as the base model is trained independently of the Hydra heads, it is not obvious whether sufficient information regarding the context is encoded in the last token's hidden state. To better aggregate relevant information over the entire context for use by the Hydra heads, we propose to extend the base LLM with an additional decoder layer which is used solely to produce better input representations for the draft model. While each Hydra head is still a single layer MLP, they each now take as input the additional decoder layer's representation of the final token in the already generated sequence. As the additional decoder layer is trained in conjunction with the Hydra heads, it can learn what information from the already generated sequence is useful for the Hydra heads. We note that all Hydra heads share the same additional decoder layer hidden state, meaning the additional decoder layer is only queried once per Hydra decoding step. We refer to the resulting Hydra head architecture consisting of an additional decoder layer along with the standard MLP as the *PrefixMLP* Hydra head architecture.

### A.2.1 Results.

To test whether adding an additional Hydra-specific decoder layer improves modelling performance, we compare decoding with our proposed PrefixMLP architecture to decoding using the standard MLP-only Hydra head (Figure 6). We find that the decoding with PrefixMLP heads improves the average acceptance length by $1.12\times$ which leads to an improvement in average decoding throughput of $1.08\times$. This result suggests that aggregating context from the generated sequence in a Hydra head aware manner improves Hydra decoding performance.

## B Discovering Performant Decoding Trees

We plot the results from searching for the optimal tree at each batch size investigated in Figure 7, Figure 8, and Figure 9 for Medusa, Hydra, and Hydra++ decoding respectively. For all decoding strategies evaluated, as the batch size increases the size of the tree which maximizes throughput decreases as well.

## C Eagle Decoding

Concurrently to our work, Li et al. (2024) have proposed the EAGLE decoding framework. Similar to existing speculative decoding techniques based on draft heads, the draft model in EAGLE leverages the base model's hidden states as input. However, EAGLE does not use a

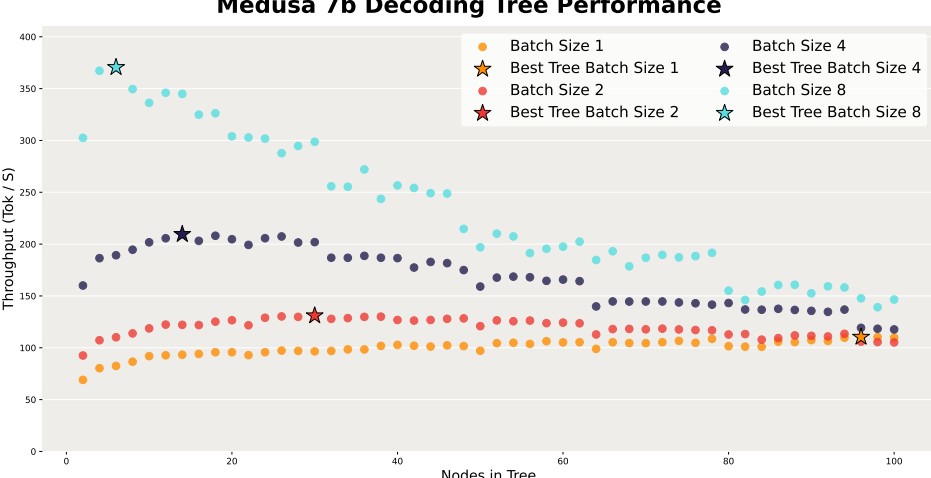

Figure 7: Results from determining the optimal decoding tree for Medusa with a 7B base model. Each point represents the throughput measured when using the tree that maximizes the expected accepted length for a given tree size. For each setting considered, the point marked by a star denotes the tree size that achieves the greatest throughput.

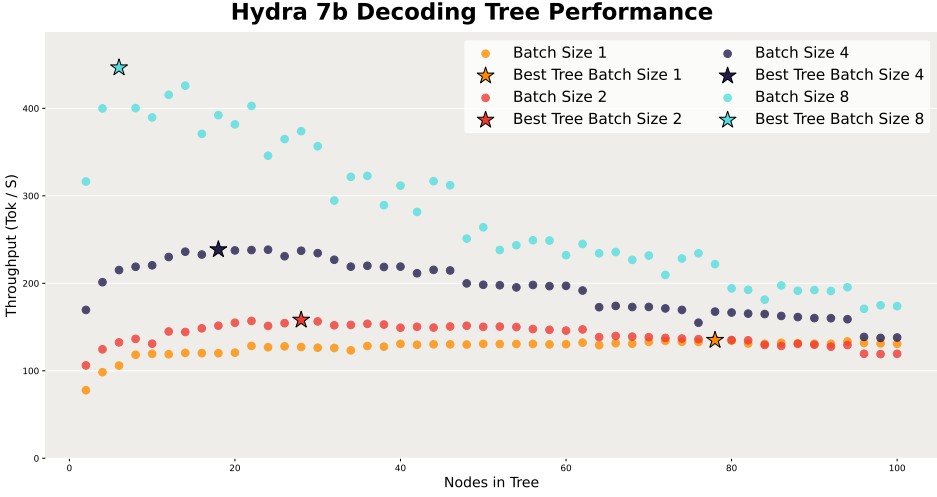

Figure 8: Results from determining the optimal decoding tree for Hydra with a 7B base model. Each point represents the throughput measured when using the tree that maximizes the expected accepted length for a given tree size. For each setting considered, the point marked by a star denotes the tree size that achieves the greatest throughput.

collection of draft heads and instead uses a singular draft head as the draft model. Similar to our work, EAGLE introduces sequential dependence to their draft head. Concretely, the EAGLE draft head is structured as a transformer decoder layer which takes as input both the hidden states and input embeddings of the entire sequence. During each step of generating a candidate continuation, the EAGLE draft model not only predicts the next token in the continuation, but also predicts an estimate of the hidden state that the base model would have computed for that candidate token. EAGLE then extends the input to its draft head with both the input embedding of the predicted token, as well as the estimated hidden state.

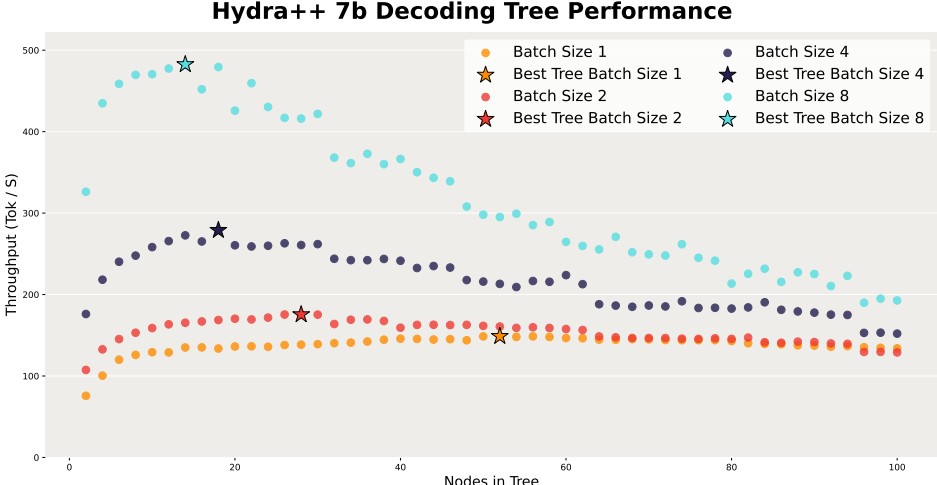

Figure 9: Results from determining the optimal decoding tree for Hydra++ with a 7B base model. Each point represents the throughput measured when using the tree that maximizes the expected accepted length for a given tree size. For each setting considered, the point marked by a star denotes the tree size that achieves the greatest throughput.

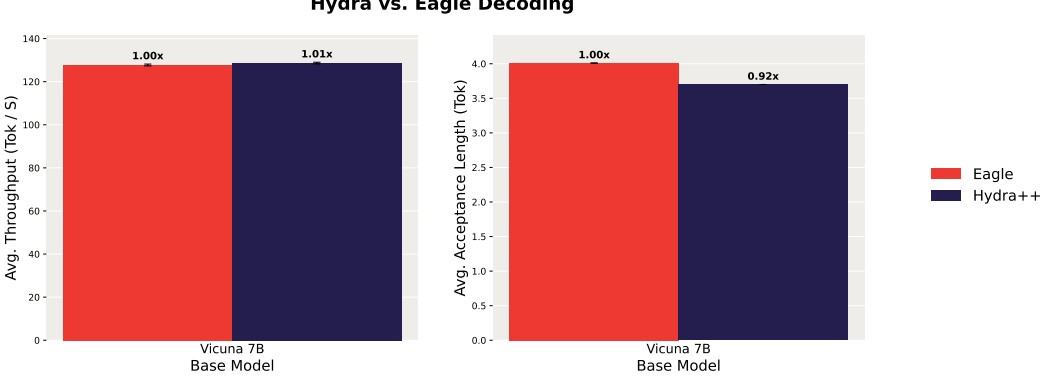

Figure 10: Comparison of Hydra++ and EAGLE. We find that both draft heads achieve comparable throughput.

Li et al. (2024) demonstrate that EAGLE decoding provides a speedup relative to Medusa's sequentially-independent draft heads. Given that EAGLE was developed entirely independently of Hydra, we believe that Hydra and EAGLE, taken together, constitute valuable evidence that the benefits of sequential dependence in speculative decoding are robust and replicable.

We independently train and evaluate EAGLE draft heads using Vicuna 7B as the base model. We compare the throughput and acceptance length of EAGLE and Hydra++ in Figure 10. We find that while EAGLE achieves a higher average acceptance length, both EAGLE and Hydra++ achieve comparable decoding throughput. We attribute this to the added overhead of EAGLE draft heads, as they require querying a full self-attention block for each position in the candidate continuation, whereas Hydra++ only queries an additional self-attention block once per decoding step, with the rest of the computation in the Hydra++ draft model being done by shallow MLPs.

| Model | Prefix Attention | Head 1 | Head 2 | Head 3 | Head 4 |
|---|---|---|---|---|---|
| Medusa | - | 0.3 | 0.3 | 0.3 | 0.3 |
| Hydra++ | 1.2 | 0.6 | 1.4 | 1.2 | 0.7 |

Table 1: Breakdown of overhead during speculative decoding in milliseconds. We report the time spent performing prefix attention and decoding for each speculative head. As can be seen, Hydra decoding incurs greater overhead than Medusa decoding. Despite the overhead, Hydra decoding leads to end-to-end throughput improvements over Medusa.

| Model | MT Chat | Translation | Summary | QA | Math | RAG | Avg. |
|---|---|---|---|---|---|---|---|
| Medusa | 2.00× | 1.68× | 1.59× | 1.67× | 1.98× | 1.54× | 1.75× |
| Hydra++ | 2.52× | 2.02× | 1.89× | 2.08× | 2.59× | 1.87× | 2.17× |

Table 2: Relative improvement in decoding throughput over standard autoregressive decoding for both Medusa and Hydra++ on the SpecBench evaluation suite. We find that across all task categories, Hydra++ achieves significantly better throughput than Medusa decoding.

## D   Analysis of Hydra Head overheads

To better understand the efficiency gains from Hydra decoding, we analyze the overhead introduced by both: 1) the additional decoder layer from prefix attention and 2) the sequential dependence of Hydra heads. Namely, to introduce sequential dependence, Hydra heads additionally take as input the embeddings of the preceding speculated tokens. As such, the first layer of a Hydra head has a greater number of input features than a Medusa head, and the number of input features grows for later Hydra heads. We report the average time spent performing prefix attention and the time spent in each speculative decoding head for both Medusa and Hydra++ at the 8B model scale with batch size one in Table 1. To contextualize the time spent performing speculative decoding, the average time to perform a decoding step through the base model is 28 milliseconds. While Hydra decoding incurs additional overhead as compared to Medusa, Hydra still achieves an end-to-end throughput improvement as it improves the average acceptance length.

## E   Evaluation on SpecBench

In addition to MT-Bench, we evaluate the performance of Hydra++ on the SpecBench (Xia et al., 2024) speculative decoding evaluation suite. While MT-Bench is a chat based benchmark, SpecBench includes a broader range of tasks to test the performance of speculative decoding methods in a variety of settings. Specifically, SpecBench is composed of multi-turn chat (MT Chat), translation, summary, QA, math, and retrieval augmented generation (RAG) tasks. We report the relative improvement in throughput over autoregressive decoding for both Medusa and Hydra++ 8B at batch size one in Table 2.

We find that Hydra++ significantly outperforms Medusa across all tasks in SpecBench. Averaged across all tasks, Hydra++ achieves an improvement in throughout over Medusa of 1.24×. The speedup we observe over Medusa on SpecBench is very similar to the speedup measured on MT-Bench (1.27×), suggesting that the gains from Hydra++ relative to Medusa generalize to a broad range of tasks. For both Medusa and Hydra++, the summary and RAG tasks see the smallest improvement in throughput over the baseline. An important area for further work is to investigate whether the speedup on these tasks can be improved from increasing the amount of summarization and RAG data seen during training, or whether the reduced performance from speculative decoding is inherent to the task.

