# OpenReview forum: "Hydra: Sequentially-Dependent Draft Heads for Medusa Decoding"
_colmweb.org/COLM/2024/Conference — COLM_

### Official Review · Reviewer_71U9 · 2024-05-05

**Rating:** 7
**Confidence:** 3
**Ethics Flag:** 1

**Summary:**

This paper works on improving the drafting accuracy of speculative decoding. Authors mention a important problem that in the Medusa like speculative decoding methods, the draft heads make predictions without considering earlier tokens in the current candidate continuation. To solve this problem, they propose a drop-in sequentially dependent method to standard draft heads. This method increases the average candidate continuation acceptance length and further accelerates LLM inference.

**Questions To Authors:**

A suggestion:  The task influences the speculative decoding method a lot, but the paper only evaluates on several benchmarks. According to this leaderboard (https://github.com/hemingkx/Spec-Bench), Hydra achieves a good performance on a wide range of tasks. It would be beneficial if the authors could provide additional experimental results to demonstrate the effectiveness of the method across different tasks.

**Reasons To Accept:**

1. The motivation is clear. The method is consistent with the motivation. The sequentially dependent method is novel and naturally considers earlier tokens in the current candidate continuation.

2. The experiments demonstrate that the proposed method significantly outperforms the baselines (2.70× acceleration).

3. Authors explore a lot on the architecture and training objective (section 3.1) . I especially appreciate the detailed analysis and discussions on this method.

**Reasons To Reject:**

The paper mainly compares with Medusa but lacks comparison with other speculative decoding methods. [1][2]

[1] Li, Yuhui, et al. "Eagle: Speculative sampling requires rethinking feature uncertainty." arXiv preprint arXiv:2401.15077 (2024).

[2] Chen, Charlie, et al. "Accelerating large language model decoding with speculative sampling." arXiv preprint arXiv:2302.01318 (2023).

---

> ### Author Rebuttal · Authors · 2024-05-30
>
> # Could we compare to Eagle and Speculative Sampling?
>
> Although Eagle [1] is concurrent work to ours, we include a comparison with Eagle in our related work section and in Appendix C of our submission.
>
> Following your suggestion to compare with Speculative Sampling [2], we have evaluated Speculative Sampling, Medusa, and our Hydra++ model under the SpecBench [3] evaluation harness, the results of which we present in the next subsection.
>
> # Could we evaluate Hydra on more benchmarks?
>
> We thank reviewer 71U9 for the suggestion to evaluate on SpecBench. We have benchmarked our 7B Medusa baseline and Hydra++ model, as well as Speculative Sampling [2], on the SpecBench [3] benchmark. We find that the relative gains from Hydra++ over Medusa generalize to this larger benchmark, and that Hydra compares favorably to Speculative Sampling [2]. We would be happy to include these results in the final paper.
>
> **Throughput relative to non-speculative decoding by SpecBench task:**
> |Method|Multi-Turn Conversation|Translation|Summarization|QA|Math Reasoning|RAG|Overall|
> |-|-|-|-|-|-|-|-|
> |Speculative Sampling|1.66×|1.13×|1.62×|1.49×|1.46×|1.54×|1.49×|
> |Medusa|2.00×|1.68×|1.59×|1.67×|1.98×|1.54×|1.75×|
> |Hydra++|2.52×|2.02×|1.89×|2.08×|2.59×|1.87×|2.17×|
>
> [1] Li, Yuhui, et al. "Eagle: Speculative sampling requires rethinking feature uncertainty." arXiv:2401.15077 (2024).
>
> [2] Chen, Charlie, et al. "Accelerating large language model decoding with speculative sampling." arXiv:2302.01318 (2023).
>
> [3] Xia, Heming, et al. "Unlocking efficiency in large language model inference: A comprehensive survey of speculative decoding." arXiv:2401.07851 (2024).

---

### Official Review · Reviewer_YfQn · 2024-05-12

**Rating:** 8
**Confidence:** 4
**Ethics Flag:** 1

**Summary:**

This paper proposed a new speculative decoding scheme based on Medusa. This is different from Medusa which only considers the LLM's hidden state as the input and predicts the future tokens, this paper proposed to take the hidden stated and predicted candidates as input, which enhances the "dependencies" rather than fully factorized. Additionally, authors explored coupled improved model architecture and the training objectives,  leading to the "Hydra++". Empirical results showed a two-digit margin increase w.r.t baselines regarding averaged accepted drafts, and throughput. In general, I think this is an elegant idea and tailored improvements to the model and objectives showed consistent gains.

**Questions To Authors:**

- Will the different sampling methods affect the gain over acceptance?

- What are the other benchmarks that might be a good fit instead of MT-bench?

**Reasons To Accept:**

- The idea itself is well-motivated and somewhat elegant to me. The predicted tokens are at hand information and could be used. It feels like adding an "autoregressiveness" in the draft heads.  I would argue the idea is simple yet effective.

- Authors also explored the design space of the model architectures, sampling methods, and training objectives. The design choices are well illustrated and motivated. And the final "hydra++" is pretty well-tuned.

- The evaluation is comprehensive and empirical results are showing consistent gains even w.r.t. strong baselines.

Additionally, the paper is pretty easy to follow and mathematical explanations are clear.

In general, this paper is clear and solid with well-justified motivation and clean techniques. Empirical results also demonstrated gains.

**Reasons To Reject:**

- Sample methods: I would encourage authors to consider more sampling methods, for example, top-p.
- Related work might consider adding some concurrent work?

---

> ### Author Rebuttal · Authors · 2024-05-30
>
> # What about other sampling strategies (e.g. top-p)?
>
> We would be happy to include throughput results for stochastic sampling using other sampling strategies (e.g. top-p) in the final paper.
>
> We note that in section 6.3 we already investigate "typical-acceptance" sampling, which is non-greedy and stochastic. We find that Hydra++ achieves commensurate quality to temperature sampling from the base model, while maintaining high candidate acceptance lengths. We present results for typical sampling in Figure 4 of our submission.
>
> # Could we add concurrent work to the related work section?
>
> The three concurrent works we are aware of are [1], [2], and [3]. We already include a comparison to [1] in the related work section and Appendix C of our submission. The preprints [2] and [3] were released 2 weeks before and 4 weeks after the final COLM submission deadline, respectively. We would be happy to include references to these preprints in our related work section.
>
> [1] Li, Yuhui, et al. "Eagle: Speculative sampling requires rethinking feature uncertainty." arXiv:2401.15077 (2024).
>
> [2] Zhang, Aonan, et al. "Recurrent drafter for fast speculative decoding in large language models." arXiv:2403.09919 (2024).
>
> [3] Wertheimer, Davis, et al. "Accelerating Production LLMs with Combined Token/Embedding Speculators." arXiv:2404.19124 (2024).
>
> # Could we evaluate on benchmarks beyond MT-bench?
>
> We thank reviewer YfQn for the suggestion. In response to your review, we have evaluated Hydra++ on the multi-task SpecBench benchmark [4]. SpecBench includes multi-turn conversation, summarization, RAG, translation, question-answering, and mathematical reasoning tasks. We evaluated our 7B Medusa and Hydra++ models on SpecBench, as well as a Speculative Sampling [5] baseline. We find that the relative gains from Hydra++ over Medusa generalize to SpecBench:
>
> **Throughput relative to non-speculative decoding by SpecBench task:**
> |Method|Multi-Turn Conversation|Translation|Summarization|QA|Math Reasoning|RAG|Overall|
> |-|-|-|-|-|-|-|-|
> |Speculative Sampling|1.66×|1.13×|1.62×|1.49×|1.46×|1.54×|1.49×|
> |Medusa|2.00×|1.68×|1.59×|1.67×|1.98×|1.54×|1.75×|
> |Hydra++|2.52×|2.02×|1.89×|2.08×|2.59×|1.87×|2.17×|
>
> [4] Xia, Heming, et al. "Unlocking efficiency in large language model inference: A comprehensive survey of speculative decoding." arXiv:2401.07851 (2024).
>
> [5] Chen, Charlie, et al. "Accelerating large language model decoding with speculative sampling." arXiv:2302.01318 (2023).

---

### Official Review · Reviewer_ngjH · 2024-05-13

**Rating:** 7
**Confidence:** 3
**Ethics Flag:** 1

**Summary:**

The paper introduces Hydra heads to introduce sequential dependency for the draft heads based on Medusa decoding. The paper conducts experiments using the Vicuna family of models to evaluate the throughput and token acceptance length compared to autoregressive decoding and Medusa and shows that the proposed method can improve on both metrics. The paper also investigates other aspects of the proposed Hydra methods by incorporating some recent techniques such as teacher loss and PrefixMLP to improve the framework further and thus presents Hydra++, which shows better performance with the combination of these techniques.

**Questions To Authors:**

1. See the weakness above.
2. For Figures 2 and 3, why is there a confidence interval plotted (if I understand correctly) for the Hydra only on Avg. Throughput results? Please correct me if I miss anything.
3. Typo: the word "using" is repeated twice in the 2nd line of section 7.

**Reasons To Accept:**

1. The presentation is clear and easy to understand. The motivation for incorporating sequential dependence to Medusa decoding is simple but seems effective from the experiments.
2. The studied problem is important and impactful for current LLM practices for decoding efficiencies.
3. The exploration of other techniques is quite extensive seeking alternative implementations of Hydra heads. The results and observations can provide useful insights for practitioners.

**Reasons To Reject:**

1. It would be good to have a cost and efficiency analysis for the proposed method since taking the previous tokens to the draft heads seems to scale with the number of heads.
2. The optimal draft head recipe section is not comprehensive enough to understand the intuitions behind some observations. For example, could the authors elaborate more on why adding noise to Hydra did not improve the performance of the models? The motivation to add noise here is unclear since the referenced paper is about using noisy embeddings for instruction tuning. It would be better to see if adding noise to the speculative decoding generally fails or if there are other observations for the relationship between noisy embeddings and decoding performance.
3. Other than greedy decoding, Some results on other temperatures for evaluation would be interesting to see how randomness would affect the observations.

---

> ### Author Rebuttal · Authors · 2024-05-30
>
> # Could we include a cost analysis for the overhead of using Hydra draft heads?
>
> We thank reviewer ngjH for the suggestion, and agree that including a discussion of the overhead of running the draft model would improve the paper. We have measured the overhead of each draft head in our 7B Medusa and Hydra++ models on MT-Bench, and obtained these results:
>
> **Absolute time (milliseconds) spent running draft heads per decoding step:**
> | Model | Extra Decoder Layer | Head 1 | Head 2 | Head 3 | Head 4 |
> |-|-|-|-|-|-|
> |Medusa|-|0.3|0.3|0.3|0.3|
> |Hydra++|1.2|0.6|1.4|1.2|0.7|
>
> For reference, a single forward pass through the base model during the verification step takes 28 milliseconds.
>
> One might expect that later Hydra++ heads would always have more overhead than earlier heads, because their input dimension is larger. However, later layers of our speculation tree can include fewer nodes than earlier layers, providing a counter-balancing effect on run time.
>
> # What is the intuition behind e.g. adding noise?
>
> Inspired by the Neftune paper, our original motivation for adding noise to the embeddings was as a form of data augmentation / regularization, to allow us to train for more epochs without overfitting. Although the Neftune paper focused on the instruction-tuning setting, our hope was that the regularization benefits it found might apply to use cases beyond instruction-tuning. We would like to clarify that we only experimented with adding noise to embeddings at training time, not during inference. We will make all of these points more clear in our camera-ready version.
>
> If there are any other aspects of our draft head recipe experiments which the reviewers would like us to clarify, we would be happy to address them.
>
> # What about sampling with higher temperatures?
>
> We would be happy to include throughput results for temperature sampling at nonzero temperatures in the final paper.
>
> We note that in section 6.3 we already investigate "typical-acceptance" sampling, which is non-greedy and stochastic. We find that Hydra++ achieves commensurate quality to temperature sampling from the base model, while maintaining high candidate acceptance lengths. We present results for typical sampling in Figure 4 of our submission.
>
> # Where are the confidence intervals in Figures 2 and 3?
>
> We plot confidence intervals for all throughput results in the paper, although in some cases the error is so small that they can be difficult to see.

---

> > ### Comment · Reviewer_ngjH · 2024-06-04
> > **Thank you for the response**
> >
> > Thanks to the authors for their responses. I think the new results and explanations are reasonable and expected. I would like to maintain my current evaluations.

---

### Decision · Program_Chairs · 2024-07-10

**Decision:**

Accept

**Comment:**

The paper proposes a technique (Hydra) to train multiple prediction heads in an autoregressive language models to speed up speculative decoding. By having different heads depend on each other, the proposed technique improves on Medusa (where the heads are independent). Careful choice of the training objective and head architectures yield up to 1.3x compared to Medusa (and 2.7x compared to standard decoding).
+ The design is well-motivated and elegant.
+ Strong empirical results on speculative decoding speedup.
Overall the paper is a valuable contributed to the area of speculative decoding.